# Anaphylactic Reactions Due to *Triatoma protracta* (Hemiptera, Reduviidae, Triatominae) and Invasion into a Home in Northern California, USA

**DOI:** 10.3390/insects12111018

**Published:** 2021-11-12

**Authors:** Norman L. Beatty, Zoe S. White, Chanakya R. Bhosale, Kristen Wilson, Anthony P. Cannella, Tanise Stenn, Nathan Burkett-Cadena, Samantha M. Wisely

**Affiliations:** 1Department of Medicine, Division of Infectious Diseases and Global Medicine, University of Florida College of Medicine, Gainesville, FL 32610, USA; 2Emerging Pathogens Institute, University of Florida, Gainesville, FL 32610, USA; wisely@ufl.edu; 3Department of Wildlife Ecology and Conservation, University of Florida Institute of Food and Agricultural Sciences, Gainesville, FL 32610, USA; zseganish@ufl.edu (Z.S.W.); cbhosale@ufl.edu (C.R.B.); knwilson@ufl.edu (K.W.); 4Department of Medicine, Division of Infectious Diseases and International Medicine, University of South Florida Morsani College of Medicine, Tampa, FL 33610, USA; acannella@usf.edu; 5Entomology and Nematology Department, Florida Medical Entomology Laboratory, University of Florida, Vero Beach, FL 32962, USA; tanise@ufl.edu (T.S.); nburkettcadena@ufl.edu (N.B.-C.)

**Keywords:** kissing bug bites, triatomine, anaphylaxis, *Triatoma protracta*, Chagas disease, *Trypanosoma cruzi*, California, United States

## Abstract

**Simple Summary:**

Kissing bugs are bloodsucking insects found throughout the Western Hemisphere, including the United States, but also within certain regions of the Western Pacific, India, the Middle East, and Africa. Within the Americas, these insects are known to harbor a parasite known as *Trypanosoma cruzi*, the causative agent of an infection in humans and other mammals known as Chagas disease. The infection can be spread through the fecal matter of the kissing bug when exposed to the skin or ingested from contaminated food or drink products. Kissing bugs will invade human homes and bite residents and their pets. The bite from a kissing bug can also lead to serious allergic reactions, including anaphylaxis. A potentially life-threatening allergic response typically needs emergency medical attention. We describe a home that was invaded by kissing bugs in northern California where the resident developed serious allergic reactions to the bite. The kissing bugs were identified and a blood meal investigation found the presence of human blood as well as the parasite, *Trypanosoma cruzi*. The resident was tested extensively for chronic Chagas disease due to his repeated exposure to the kissing bug but was found to not have the disease. Those who live in regions where kissing bugs are found naturally should be aware that their bites can be highly allergenic.

**Abstract:**

Background: *Triatoma protracta* is a triatomine found naturally throughout many regions of California and has been shown to invade human dwellings and bite residents. A man living in Mendocino County, California, reported developing anaphylactic reactions due to the bite of an “unusual bug”, which he had found in his home for several years. Methods: We conducted environmental, entomological, and clinical investigations to examine the risk for kissing bug invasion, presence of *Trypanosoma cruzi*, and concerns for Chagas disease at this human dwelling with triatomine invasion. Results: Home assessment revealed several risk factors for triatomine invasion, which includes pack rat infestation, above-ground wooden plank floor without a concrete foundation, canine living in the home, and lack of residual insecticide use. Triatomines were all identified as *Triatoma protracta*. Midgut molecular analysis of the collected triatomines revealed the detection of *T. cruzi* discrete typing unit I among one of the kissing bugs. Blood meal PCR-based analysis showed these triatomines had bitten humans, canine and unidentified snake species. The patient was tested for chronic Chagas disease utilizing rapid diagnostic testing and laboratory serological testing, and all were negative. Conclusions: *Triatoma protracta* is known to invade human dwellings in the western portions of the United States. This is the first report of *T. cruzi*-infected triatomines invading homes in Mendocino County, California. *Triatoma protracta* is a known vector responsible for autochthonous Chagas disease within the United States, and their bites can also trigger serious systemic allergic reactions, such as anaphylaxis.

## 1. Introduction

In the United States (US), there are 11 kissing bug species are found to naturally occur in various regions of at least 29 states spanning the southern half of the country [1,2]. Autochthonous Chagas disease (CD) has been reported in the US with increasing awareness and testing due to kissing bug exposures [2]. Four triatomine species are well known to invade human dwellings in the US—namely, *Triatoma sanguisuga*, *T. protracta*, *T. rubida*, and *T. gerstaeckeri* [2,3,4,5,6,7]. Being hematophagous insects, they will take a blood meal from many hosts, including human beings. During the process of blood meal acquisition, kissing bugs will inject several substances into the bite site, including salivary proteins, which are mainly responsible for anticoagulation effects and dampening pain signals [8,9]. Kissing bug bites can cause cutaneous allergic reactions and even systemic manifestations such as anaphylaxis [8,9]. At least one death has been reported in Arizona of an individual who experienced anaphylaxis from a kissing bug bite [10]. Currently, there are no immune desensitization therapies available for those at risk for severe or life-threatening reactions to kissing bug bites.

A 37-year-old male with a past medical history of well-controlled HIV disease on antiretroviral therapy (ART) presented to an outpatient infectious diseases clinic to establish care after moving from northern California approximately three months ago. The patient had been adherent to ART, and a recent HIV-1 RNA quantitative assay in the blood was undetectable (<20 copies/mL), with CD4 count at 920 cells/UL. After review of HIV disease history, the patient endorsed a recent evaluation in an emergency department in northern California approximately four months ago with intense whole body pruritis, hives, abdominal pain, swelling of the face and tongue, tachycardia, and low blood pressure after sustaining a bite from what he believed was a “kissing bug”. He was admitted to the hospital and was treated with oral antihistamines, intravenous glucocorticoids, and intramuscular epinephrine but did not require the intensive care unit or intubation for airway protection. He was discharged after three days with a diagnosis of anaphylactic reactions with angioedema due to an insect bite and given a prescription for epinephrine autoinjector. The patient had no other known allergies to medications or other insect bites or stings. The patient reported having suffered “hundreds” of bites from what he believed were “kissing bugs” over a 10-year period. Intermittently, he would notice the presence of “smaller” kissing bugs, which were determined to be nymphal stages after positively identifying them when being presented with photographs of local kissing bugs. These bites 10 years ago initially started with local erythematous welts that became painful and persisted for several weeks until self-resolving. As the years passed, he continued to find the bugs inside his home and continued to sustain similar bites. He did not sleep under a mosquito net. He began to experience systemic symptoms with each bite, including generalized pruritis and diffuse cutaneous rashes. He would often take over-the-counter antihistamines to help with these symptoms, but these practices had stopped working for the past year. This latest episode was his first experience with such a severe reaction requiring hospitalization and treatment of anaphylaxis with angioedema.

## 2. Materials and Methods

Assessment of home—The patient lived in a log cabin structure that was built in the 1980s. The home is located in Mendocino County, California (Figure 1), in a rural setting within the Mendocino National Forest, approximately 2500 feet (~760 m) above sea level. It is a single floor structure that is constructed entirely of wood and has an above-the-ground wood plank floor without a concrete foundation. The home is surrounded by various coniferous trees native to the region. Evidence of pack rat nesting is around the home near certain naturally occurring rock formations. The resident of the home reported that no insecticides are used outside or inside the dwelling. Baiting/trapping and removing pack rat nests from around the structure has been attempted previously. The surrounding landscape is well groomed without a clutter of trash or unused items. One healthy canine is kept indoors. No other domestic or farm animals are kept on the property.

Phenotypic and molecular identification of triatomines—Six insects collected from inside the home were morphologically assessed using entomological keys [11]. Legs (N = 2) of each bug were removed and used for DNA extraction and conventional PCR assays targeting the ITS gene for amplification [12]. These PCR products underwent Sanger sequencing, cleaned, and subjected to GenBank Basic Local Search Alignment Tool (BLAST) inquiry.

Triatomine blood meal analysis—The midgut of each individual bug was dissected and triturated in NaCl then heated in extraction solution (InstaGene matrix). The total DNA extracted underwent PCR assays targeting the mitochondrial and ribosomal genes to amplify conserved regions of host DNA, following published protocols [13]. Samples amplified by PCR were Sanger sequenced and subjected to library search via GenBank BLAST. Host species were considered a positive match when query sequences have >95% similarity to reference sequences.

Molecular detection of *T. cruzi* and discrete typing unit (DTU) determination—We dissected midgut material from the abdomen and rectum for six individual *T. protracta* kissing bugs. Gut contents were placed in 600 µL of QIAGEN Gentra Purgene (QIAGEN, Valencia, CA, USA) lysis buffer and 5 µL Proteinase K solution (20 mg/mL) and incubated at 56 °C overnight shaking at 350 RPM. If the material was not fully digested, we performed a second overnight digest with additional Proteinase K. We then extracted genomic DNA following the manufacturer’s protocol the following day. Slight modifications included a 5 min protein precipitation at 4 °C and one additional ethanol wash. Samples were eluted into 100 µL of TE buffer and stored at −20 °C. DNA concentrations varied from 8.6 to 149 ng/µL. To detect genomic DNA of *T. cruzi*, we used conventional PCR with primer set, TCZ1, and TCZ2, which amplified a 188 bp satellite repeat region [14]. We ran each set of assays with a positive control that contained 10 ng/µL of DNA from *T. cruzi* DTU I grown in cell culture and a negative control with molecular grade water instead of template DNA. After amplification, gel electrophoresis was performed to identify samples that were positive for *T. cruzi*. We confirmed the identity of the DNA fragments by Sanger sequencing (Eurofins Genomics, Louisville, KY, USA). Samples that were confirmed positive were subsequently typed for DTU using a stepwise multiplex real-time PCR [15]. Samples were amplified using an ABI 7500 Fast Real-Time PCR System Machine (Thermofisher, Waltham, MA, USA). For this assay, DTU specific constructs for each primer-probe gene target were used as positive controls. A known negative *T. cruzi* gDNA sample was used as a template control, and molecular grade water was used as a negative, no template, control. Samples were considered positive if the real-time assay had a C_T_ value < 38 and negative if it was ≥39 cycles.

Chagas disease testing of individual bitten by triatomines—The homeowner underwent an investigation for chronic Chagas disease. A complete review of systems and physical exam was performed in the clinic. He completed a chest X-ray and a 12-lead electrocardiogram. Rapid diagnostic testing with two lateral flow assays (Chagas Detect™ Plus; InBios International, Inc., Seattle, WA, USA; DPP^®^ Chagas System; Chembio Diagnostic Systems, Inc., Hauppauge, NY, USA) was performed in the clinic following each manufacturer’s instructions. Serum samples were sent to three other references labs (Quest Diagnostics laboratory; Associated Regional and University Pathologists Laboratory, Salt Lake City, UT, USA; Viracor Eurofins Clinical Diagnostics, Lee’s Summit, MO, USA) to further investigate for the presence of anti-*T. cruzi* IgG antibodies.

## 3. Results

All six triatomines were confirmed to be *T. protracta* (Uhler) morphologically using entomological keys [11]. Further molecular confirmation was conducted using the internal transcribed spacer two gene sequences for *T. protracta* (GenBank Accession Numbers: OK663059-OK663064), and both were a positive match at 97.2–100% to reference sequences (GenBank Accession Numbers: JX872263.1, JQ282715.1, JQ282714.1, JQ282713.1) for *T. protracta* after Sanger sequencing and GenBank BLAST inquiry. Five were adult females (Figure 2) and one was an adult male. Our bloodmeal analysis revealed that human blood was found in five out of the six of these triatomines. Domestic canine and an undetermined snake species (<95% match to multiple species) were found in one other triatomine each. Two triatomines were positive for *T. cruzi* by conventional PCR and confirmed by Sanger sequencing. *Trypanosoma cruzi* DNA sequences generated in this study (GenBank Accession Numbers: OK586820, OK586821) had a 99.7% similarity to other *T. cruzi* DTU I sample (GenBank Accession Numbers: LT220301, LT220305) previously collected in Texas. One specimen was positive for DTU I with a C_T_ value of 36, and the other was inconclusive for a DTU strain. Our investigation of Chagas disease transmission for this homeowner was centered on possible chronic Chagas disease given that our patient was currently asymptomatic and greater than three months from the last exposure to the kissing bug. A complete review of systems and physical exam did not reveal any abnormalities. The 12-lead electrocardiogram was normal in the clinic and a chest X-ray did not show any abnormalities. The two rapid diagnostic tests for Chagas disease (Chagas Detect™ Plus; DPP^®^ Chagas System) were negative in the clinic (Figure 3). All three anti-*T. cruzi* IgG serological assays (Hemagen Chagas kit EIA, Ortho *T. cruzi* ELISA, Weiner Chagatest ELISA recombinante v.3.0) were non-reactive.

## 4. Discussion

Our patient developed severe allergic reactions to the bites of *T. protracta* after continued invasion into his home situated in Mendocino County located in northern California. Triatomine bites commonly cause an intense local cutaneous reaction, such as erythematous welt at the bite site that can persist for weeks and are commonly painful afterward [8]. As demonstrated in our patient, repeated bites in some can lead to the development of anaphylactic reactions [10,16,17]. This has been described in Arizona with those bitten by *T. rubida*, *T. protracta*, and *T. recurva* [5,16]. Among one cohort study in Arizona bitten by triatomines in the Sonoran Desert, 10.5% (N = 11/105) experienced anaphylaxis. Other North American triatomines, such as *T. rubrofasciata* (invasive), *T. gerstaeckeri*, and *T. sanguisuga* have also been shown to cause serious allergic reactions after repeated bites [17,18]. *Triatoma protracta* has demonstrated that it will invade human dwellings in California and has led to serious allergic reactions, including anaphylaxis [19]. Our investigation has revealed the first documented report of *T. cruzi* in Mendocino County, California, and *T. cruzi*-infected triatomine invasion into a human dwelling in this region of California [20,21].

Management of *Triatoma* bite allergy is primarily centered on the avoidance of future bites from this insect [5,22]. Currently, there are no commercially available treatments, but *Triatoma* bite allergen immunotherapy has been successfully performed, first attempted in 1982 [22,23]. Utilizing *T. protracta* salivary gland extract for immunotherapy in those with severe allergic reactions has been shown to significantly decrease reactions when later challenged with a future bite [22,23]. The largest immunotherapy trial, to date, is from 1984, at which time five patients with known anaphylaxis from *T. protracta* bites underwent *T. protracta* salivary gland extract immunotherapy [24]. All five patients in this study had demonstrated successful immunotherapy for *T. protracta*-induced anaphylaxis and prevention of anaphylactic reactions with provocative challenges [24]. For our patient, who continued to travel back to his home in Mendocino County, we recommended that he sleep with an epinephrine autoinjector and consider sleeping under a mosquito net. Due to the known presence of *T. sanguisuga* here in northern Florida [25,26], we also recommended that the patient should sleep with an epinephrine autoinjector because he may have a similar reaction to the bite of *T. sanguisuga*. The patient was given instructions on preventing kissing bug invasion into his home in California and Florida. This includes sealing windows and doors from cracks, using screens on windows, removing animal nests (such as pack rat and opossum) and any other clutter near the home, sleeping under a mosquito net, and contacting a pest controller for residual insecticide spraying.

Autochthonous CD has been described in the western portions of the US where *T. protracta* resides naturally, including in the state of California and Arizona [27,28]. Despite sustaining “hundreds” of bites from triatomines over a 10-year period, frequently finding them inside his home, and evidence of *T. cruzi*-infected triatomines collected from inside the home, our patient did not have clinical or serological evidence of chronic CD. There has been some concern that certain *T. cruzi* DTU strains found within certain geographic regions in Mexico, as well as Central and South America, have discordant serological testing results [29,30,31,32]. The sensitivity and specificity of our available CD serological tests have not been assessed among those with autochthonous CD in the US at this time. Thus, we tested our patient with all four FDA-approved or -cleared assays for the diagnosis of CD [1], which were all negative, including another rapid diagnostic test (Chembio DPP^®^ Chagas System) that is used only in the US for research purposes currently but is frequently utilized for Chagas diagnostic purposes in other regions of Latin America. Our patient did have an HIV infection and was adherent to ART with sustained viral suppression and normal CD4 counts. Chagas disease among those with uncontrolled HIV/AIDS can lead to significant reactivation of *T. cruzi* and often times will be fatal [32,33]. Despite having CD4 counts greater than 200, *T. cruzi* DNA can still be detected in the blood of those infected with HIV, which likely indicates those living with HIV are at continued risk for reactivation Chagas disease [34]. In our patient, he likely did not have CD given the extensive serological testing that was carried out but in similar circumstances, it is advisable to screen those at-risk for CD in this population. This is also the first report of an individual living with HIV disease and the development of severe allergic reactions to triatomine bites. It is unclear if having HIV infection is contributing to this phenomenon, and more research is needed to investigate this clinical scenario.

## 5. Conclusions

Repeated bites from *T. protracta* kissing bugs can lead to serious allergic reactions. *Triatoma protracta* has been implicated in the vector-borne transmission of CD in California among several well documented cases in the US, but a majority of autochthonous cases are not well understood. In Arizona, it has been shown that CD is likely not being transmitted among those bitten by native triatomines or thought to be a rare event. However, more data are needed among those who have an invasion of triatomines into their homes in other regions of the US to confirm CD is not being transmitted among other triatomine species. Those at risk for triatomine invasion into the home and severe allergy to their bites should take precautions as anaphylaxis can be serious and even fatal.

## Figures and Tables

**Figure 1 insects-12-01018-f001:**
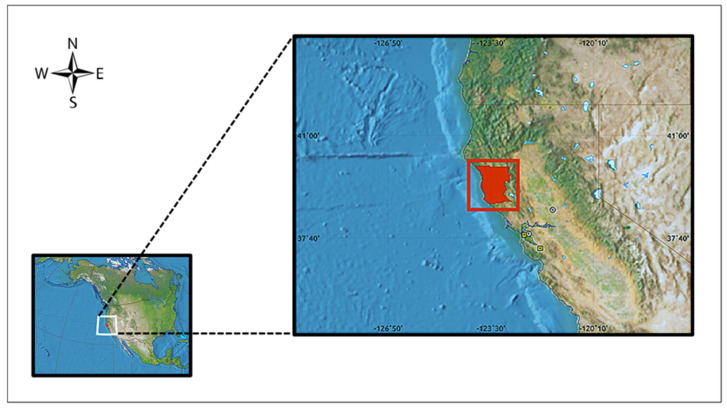
Location of Mendocino County (red square), which is situated in northern California and approximately 160 miles (257 km) north of the city of San Francisco and 280 miles (450 km) south of the state of Oregon.

**Figure 2 insects-12-01018-f002:**
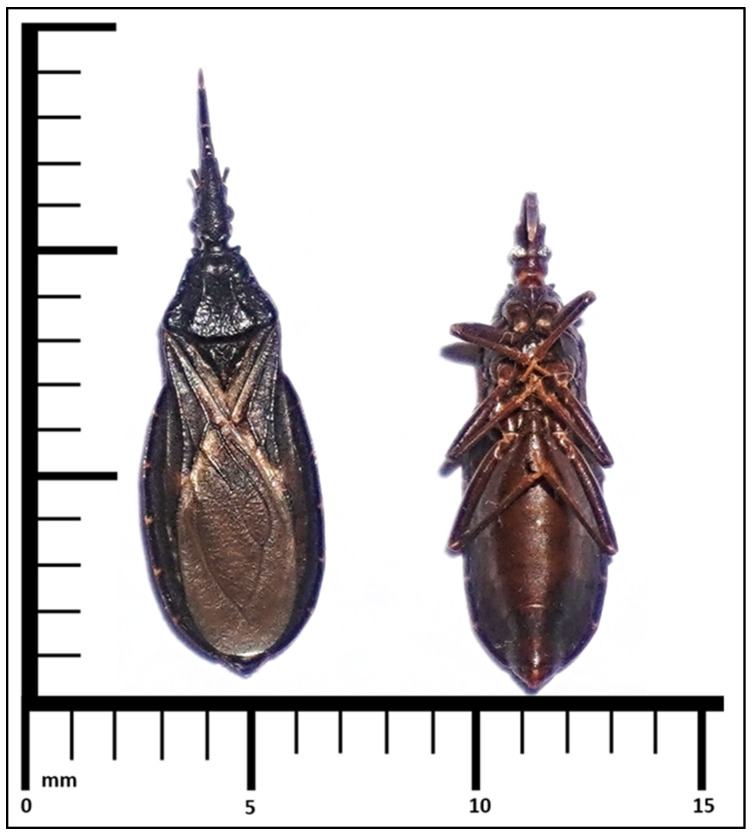
Dorsal (**left**) and ventral (**right**) views of adult female *Triatoma protracta* (Uhler) collected from home in Mendocino County, CA, USA.

**Figure 3 insects-12-01018-f003:**
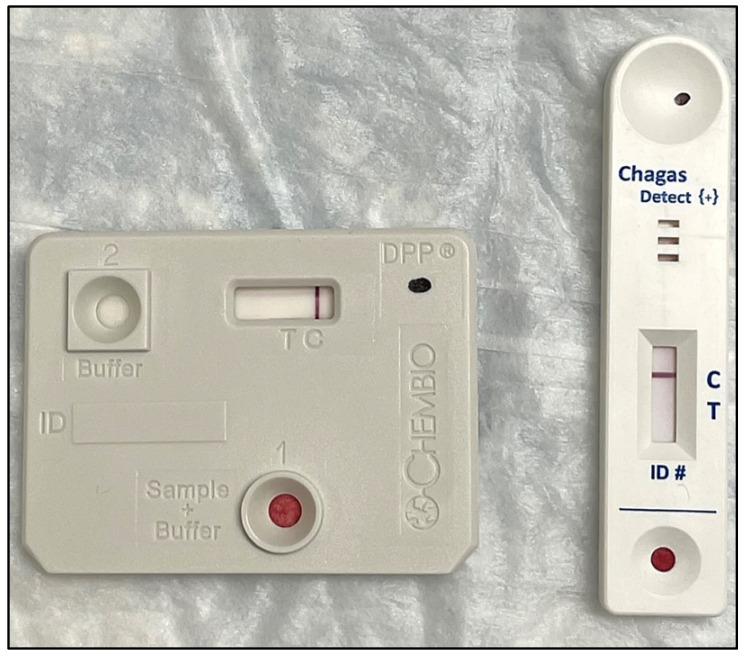
Rapid diagnostic tests for Chagas disease (**left**—DPP^®^ Chagas System; **right**—Chagas Detect™ Plus) performed in the clinic revealed negative results (T with absent line in both tests).

## Data Availability

Data are available upon reasonable request.

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
