# Peer review of "Anaphylactic Reactions Due to *Triatoma protracta* (Hemiptera, Reduviidae, Triatominae) and Invasion into a Home in Northern California, USA"

_insects, 2021, doi:10.3390/insects12111018_

Round 1

Reviewer 1 Report

  • Title: Anaphylactic Reactions due to Kissing Bug (Triatoma protracta) Bites and Invasion into a Home in Northern California, USA.

I suggest: Anaphylactic Reactions due to Triatoma protracta (Hemiptera, Reduviidae, Triatominae) and Invasion into a Home in Northern California, USA.

Please see detailed comments in attachment.

Reviewer 2 Report

The goal of this article was to report anaphylactic reactions due to triatomine bites. This is a well-written report where  the authors reviewed  the scarce literature about triatomine bites associated to allergic reactions . This is the first report of T. cruzi infected triatomines invading dwellings in Mendocino County, California, USA. The manuscript provides advance towards the current knowledge, making it appropriate to the “Insects” readership.

The following minor suggestions/corrections are offered to the reviewed version of the paper:

Line 17:Kissing bugs are bloodsucking insects found throughout the western hemisphere, including the United States".  There are a few species in the Old World and rewrite the sentence to clear this point. Ref. Galvão C. 2021. Taxonomy. In A. Guarneri, M. Lorenzo (eds.), Triatominae - The Biology of Chagas Disease Vectors, Entomology in Focus 5, https://doi.org/10.1007/978-3-030-64548-9_2

The paper below could be used to improve the discussion.  John E Moffitt et al (2003). Allergic reactions to Triatoma bites. Ann Allergy Asthma Immunol 91(2):122-8. doi: 10.1016/S1081-1206(10)62165-5.

Reviewer 3 Report

I have only minor comments:

General: It is not apparent from the current manuscript if the sequences derived here have been submitted to GenBank; the authors should clarify.

Line 95. Should be altitude ~760 metres, not km

Line 115. NaCl, not CL.

Line 132. Do the authors know the geographical origin of the DTU TCI control DNA?

Line 199. Species name sanguisuga should be italic font

Figure 2. Could the scale be placed vertically?
